# Effects of Molecular Weight and Guluronic Acid/Mannuronic Acid Ratio on the Rheological Behavior and Stabilizing Property of Sodium Alginate

**DOI:** 10.3390/molecules24234374

**Published:** 2019-11-29

**Authors:** Wenxiao Jiao, Wenxue Chen, Yuqi Mei, Yonghuan Yun, Boqiang Wang, Qiuping Zhong, Haiming Chen, Weijun Chen

**Affiliations:** 1College of Food Sciences & Engineering, Hainan University, 58 People Road, Haikou 570228, China; jwx709140172@163.com (W.J.); hnchwx@163.com (W.C.); myq877446337@163.com (Y.M.); yunyonghuan@foxmail.com (Y.Y.); E1178790644@163.com (B.W.); hainufood88@163.com (Q.Z.); 2Chunguang Agro-Product Processing Institute, Wenchang 571333, China

**Keywords:** sodium alginate, rheological behavior, emulsifying property

## Abstract

The aim of this study was to prepare sodium alginates (SAs) with different molecular weight and G/M ratio, and characterize their rheological behaviors and emulsifying properties. The result of Fourier transform infrared (FTIR) showed that the chemical bonds among the β-d-mannuronic acid- (M-), α-l-guluronic acid- (G-), and MG-sequential blocks in the SA chains were not changed significantly by acid treatment. Meanwhile, the molecular weight and G/M ratio of the SA exhibited drastic variation after acid modification. The result of rheological analysis suggesting that the apparent viscosity of SA reduced from 30 to 16.4 mPa.s with the increase of shear rate, reveals that SA solution belongs to pseudoplastic liquid. Also, the apparent viscosity of acid-modified SA solution dropped rapidly with the decrease of the molecular weight. The properties of emulsions stabilized by SA, SA-Ms, and commercial SAs were evaluated via the interface tensiometry and determination of the zeta potential, droplet size, creaming index (CI), and Turbiscan stability index (TSI). Compared with the SA-stabilized emulsion, the interfacial tension of the emulsion stabilized by SA-M increased with the decrease of the molecular weight reduced at the similar M/G ratio. The decrease in zeta potential and the increase in TSI of the emulsion were observed with the decrease of molecular weight, indicating that molecular weight plays an important role on the emulsifying ability of SA. In addition, the SA with low G/M ratio can form emulsions with stable and fine droplets.

## 1. Introduction

Sodium alginate (SA) is an important natural polysaccharide which is made up of (1-4)-linked β-d-mannuronic acid (M) and α-l-guluronic acid (G) monomers. According to the sequence of structure units (M and G), there are three block structures: M-, G-, and MG-sequential block structures [1]. The G-block is stiffer and more extended in chain configuration than the M-block due to a higher degree of hindered rotation around the glycosidic linkages [2]. Bearing in mind its gelling ability, stabilizing properties, and high viscosity in aqueous solution, SA and its derivatives are widely used in various areas such as the food (gelata, stabilizer, thickener, and film-forming agent) and cosmetic industry (mask shaper, toothpaste ingredient, and shampoo aid) because of their gel-forming ability [3,4]. Because of many carboxyl groups on M and G blocks, SA is sensitive to the cations, which makes it an effective biosorbent to remove heavy metals (Cd, Cr, and Cu), a cross-linker between the functional groups of SA and calcium to form hydrogels and a wall material of microcapsule to control the release of drug in pharmaceutical industry [5,6]. In addition, alginate and its modified forms serve as a water-soluble dietary fiber because they do not contribute significant nourishment or calories as they pass through the human digestive system. Therefore, alginates are also used as functional foods in the adjuvant treatment of obesity and diabetes.

The physical properties, the ability to form gels, and the strength of SA gels depend not only upon molecular weight distribution, but also on the uronic acid composition and the relative proportion of the three types of blocks, the source of calcium ions and the methods of preparation [7]. One significant approach is the use of inorganic or organic reinforcement to prepare polysaccharides and nanocomposites with different molecular weight and special structure (porous and cage), thereby resulting in improved mechanical properties [8], smooth surface [9], and strong adsorption capacity [10]. In the present study, the objective is to prepare SAs with different molecular weight and G/M ratio by acid modification and characterize their rheological behaviors and emulsifying properties. To this end, FTIR was used to identify the variation of the chemical bonds among G-G, M-M and G-M blocks in the SA chains. We further compared the uronic acid composition, G/M ratio, molecular weight and properties (apparent viscosity, interfacial tension, zeta potential, droplet size, creaming index (CI), and Turbiscan stability index (TSI)) of acid modified-SA (SA-M) with that of native SA.

## 2. Results and Discussion

### 2.1. Uronic Acid Composition and Molecular Weight Analysis

SA is made of β-d-mannuronic acid and α-l-guluronic acid by β-1,4-glycosidic linkage. Because of a higher degree of hindered rotation around the glycosidic linkage, the G-block is more extended in chain configuration than the M-block [11]. Therefore, the gel forming ability is related to the content of G- block. Penman reported that G-riched SA obtained brittle gel, while M-riched SA solution gave a low viscosity, a weak strength, and rigidity colloid [12]. As shown in Table 1, the yields of the modified sodium alginates were within 80%, the M/G ratio of native SA increased from 5.70 to 6.38 (SA-M4), and then decreased to 6.18 (SA-M8) after acid treatment, indicating that the M-G glycosidic linkages are selective hydrolyzed under acidic conditions [13,14]. In order to compare the effects of M/G ratio on the properties of SA in similar Mw, three commercial SAs were obtained and their M/G values are 5.70, 4.79, 6.01, and 3.97 for SA-C1, SA-C2, and SA-C3, respectively.

Figure 1 shows the GPC chromatograms of SAs, and the results are summarized in Table 1. SA had the highest Mw (8614 kDa), while the Mw of SA-Ms decreased with the acid treatment from 8614 to 2206, 1257, 979, 89.1, 853 kDa for SA-M1, SA-M2, SA-M4, SA-M6 and SA-M8, respectively. This result suggested that TFA could break the glycosidic bond of the SA chain and induce SA degradation, which is accordance with the previous reports [11,13]. As shown in Figure 1, the molecular weight of the commercial SAs (SA-C1, SA-C2, SA-C3) had similar distribution, and they belonged to the same magnitude (9926, 9113 and 9377 kDa).

### 2.2. FTIR Analysis

The FTIR was employed to identify the structure of samples, and the spectra are presented in Figure 2, exhibiting typical absorption bands of SA. A broad absorption vibration peak at 3434 cm^−1^ is assigned to the response of hydrogen bond O-H. Strong bonds at 1616 and 1415 cm^−1^ belong to the asymmetric and symmetric stretching vibrations of carboxylate group C=O on the polymeric backbone [15]. Both the absorbance bands at 1099 and 1035 cm^−1^ are attributed to C-O-C stretching vibrations. In addition, the anomeric region (950–750 cm^−1^) is assigned to vibration of uronic acid residues [16]. Compared with the native SA, the spectra of acid-modified SA showed no significant difference, indicating that TFA only broke the skeleton chain of SA and does not change the chemical bonds among the β-d-mannuronic acid- (M-), α-l-guluronic acid- (G-), and MG-sequential blocks in the SA chains.

### 2.3. Rheological Behavior

The highly viscous films at the oil–water interface can decrease the particle movement and coalescence effect, thus improving the stability of the emulsion [15,17]. Flow curves of each SA solution (0.5%, *w*/*w*) are shown in Figure 3. With the increase of shear rate, the apparent viscosity of SA decreased from 30 to 16.4 mPa.s, while the apparent viscosity of modified SA solution was closer to 0 mPa·s. This result indicated that molecular weight or the band-linkage played an important role in the viscosity of SA [11,18]. Furthermore, SA molecules became more ordered along the flow field and offered less resistance to overcome the Brownian motion so that the viscosity was decreased [18,19]. In addition, SA-C1 showed the highest viscosity compared to the other SAs. Also, lower values were observed in the viscosity of the SA-C2 solution when the share rate was below 600 s^−1^, as compared with that of SA-C3 solution. When the shear rate was higher than 600 s^−1^, the result was the opposite. From the slope of the curves, we can find that the slope fluctuation of SA-C2 was the lowest, indicating that the shear stability of SA-C2 solution was the highest. Therefore, it was deduced that the viscosity of SA may be affected by both Mw and M/G ratio.

### 2.4. Emulsifying Properties of SAs-stabilized Emulsion

Zeta potential as one of the important indication of charged polymers can reflect aggregation of two biopolymers [20]. When the zeta potential is greater than +30 mV or less than −30 mV, there is sufficient electrostatic repulsion to prevent the agglomerates between the particles [17]. According to previous studies, acidic polysaccharides such as pectin may find it difficult to hold a stable system only by electronic potential [21]. Zeta potentials of SA solution (0.01%, *w*/*w*) are shown in Figure 4. The zeta potentials shifted from −53.0 to −35.5 mV after acid-modification with the increase of hydrolysis time. Zeta potential of SA solution was almost twice as much as that of SA-M6, suggesting that the molecular weight of SA had significant effects on the zeta potential [22]. In addition, there was no significant difference among the zeta potentials of SA, SA-C1, SA-C2, and SA-C3, indicating that M/G ratio has little influence on the zeta potential of SA.

The zeta potential depends on the interface and the structure of the SA in the continuous phase. Furthermore, the zeta potential measurements are performed to evaluate the effect of SA on electrostatic deposition onto the CMIP interfacial film surrounding the droplets. At pH 7, the charge on the CMIP coated droplets was approximately −42.40 mV in the absence of sodium alginates, and the zeta potentials of the emulsions produced by SA-Cs were around −58.88 ± 0.20 mV. The changes on the droplets were much more negative in the secondary emulsions containing sodium alginate than that observed in the primary emulsion, thus suggesting that even at neutral pH, deprotonation of hydroxyl and carboxyl groups on sodium alginate molecules adsorbed to cationic patches on the CMIP-coated droplet surfaces, thereby forming a self-assembled interfacial bilayer [23,24,25,26,27]. In summary, the SA-Ms with low molecular weight failed to bind to CMIP like high molecular weight ones, indicating that molecular weight has an important influence on the application of SA as an emulsifier. The potential values of the emulsion prepared by the SA-Cs were similar, indicating that M/G has no significant effect on the stability of emulsion at the initial stage of emulsion.

An emulsifier is capable of absorbing at the interface between oil and water, and reducing the interfacial tension of the emulsion [28]. The effects of SAs on the interfacial tension between water and MCT were studied (Figure 5). Compared with the SA, the decrease of interfacial tension caused by SA-Ms depends on their Mw at the similar M/G ratio, which indicated that the decrease of Mw of SA may play negative effects on its emulsifying ability. The interfacial tensions of SA, SA-C1, SA-C2, and SA-C3 were 10.02, 10.94, 10.53, and 12.31 mN/m, respectively. Evidently, to some extent the interfacial tension increased gradually with the decrease of M/G ratio.

In the emulsion, the particle size is usually selected as a characterization tool and selected as a quality response parameter [29]. The particle mean diameter (*d*_4,3_ and *d*_3,2_) of all emulsions were shown in Figure 6. The mean droplet size was characterized in terms of the surface area mean diameter (*d*_3,2_) and volume mean diameter (*d*_4,3_), which were used to assess the emulsifying ability and emulsion stability, respectively. The *d*_4,3_ value of SA-stabilized emulsions first increased from 1.39 to 1.87 μm (SA-M6), and then decreased to 1.43 μm (SA-M8). The relatively constant *d_4,3_* values could be explained by the fact that enough particles absorbed to the droplet interfaces and stabilized them against flocculation [27]. Although emulsions stabilized by SA, SA-M1, and SA-M8 had almost the same *d*_4,3_ values, *d*_3,2_ value of SA-M8-stabilized emulsion was significant higher than those of the other two emulsions, indicating that the emulsifying ability of SA was better than SA-Ms. A significant increase in particle sizes was found in SA-C1- and SA-C3-stabilized emulsions, suggesting that SA with a high M/G ratio can form stable emulsion with smaller droplet size. In addition, the stability of emulsion was also evaluated by means of storage assessment (Figure 7) and Turbiscan evaluation (Figure 8). As shown in Figure 7, the freshly prepared emulsions were homogeneous, while a little serum layer appeared at the bottom of emulsions stabilized by SA-M4, SA-M6, and SA-M8 after storage for 4 days. After storage for 15 and 24 days, all the emulsions prepared by the modified-SAs showed clear serum layers. Furthermore, all SA-C-stabilized emulsions showed good physical stability. From the investigation of the emulsifying properties, we could found the emulsion stability of SA was closely associated with the viscosity and zeta potential of emulsion. It could be concluded that higher viscosity could efficiently enhance the stability of emulsion and the relative high negative charges on the surface of the oil droplets could prevent them from flocculation, as a result, increasing the static and spatial repulsion among the droplets [15,30]. After the polysaccharide was added, by changing the rheological properties of the aqueous medium and forming a protective film on the surface of the protein, the aggregation was inhibited, and the stability of the emulsion was enhanced. Combined with the results of molecular weight distribution and M/G ratio analysis, we could conclude that the differences in molecular parameters should be the primary reasons results in the differences in the emulsion stabilizing properties of these SAs. SAs with higher average molecular weight and viscosity could increase the emulsion stabilities significantly. [29,31] In other words, the molecular weight should be the primary reason for the different performance in stabilizing the emulsions.

To better characterize the destabilization of emulsion, Turbiscan was adopted to monitor the variation of the relative stability of each emulsion by transmitted light and backscattering intensity in Turbiscan stability index (TSI). Turbiscan backscatter data were plotted against sample height over time, and the deflocculation/coagulated emulsion stability was evaluated by analyzing the backscatter fingerprints obtained from Turbiscan [32,33]. The macroscopic fingerprint of the emulsion was formed at a given time, allowing us to examine the migration of lipid droplets [25]. The backscattering light intensity of emulsions was obtained (Figure 8A). When the transmitted light signal was zero, the stability of the sample can be analyzed by the curve of the backscattered light intensity over time. The left curve downward and the middle stable, which meant the emulsion did not show an increase in particle size during storage, and there was be a tendency of water evolution at the bottom of the emulsion. The stability of emulsions can be evaluated by Turbiscan stability index (TSI) parameter which takes into account all processes taking place in the sample (thickness of sediment and clear layer, process of particles settling). The higher the TSI value, the less stable the emulsion. The TSI value was calculated via the cumulative value of the backscattered or transmitted light intensity changes for each measurement compared to the previous measurement during the measurement time, reflecting the combined change in volume concentration and particle size throughout the measurement period. In comparison to SA-Ms-stabilized emulsions, the SA-Cs-stabilized emulsions showed much lower TSI values during 24 days of storage, indicating that SA-Cs-stabilized emulsions had a better physical stability (Figure 8B). The TSI of the modified SA- stabilized emulsion increased from 0 to 2.90−5.44 within 10 days, while the SA-Cs-stabilized emulsions maintained a relatively low value of 0.43–0.70 over 10 days. And it can be seen from Figure 8B, the TSI of the emulsion prepared by SA-M increased with the acid treatment time. It can be concluded that the molecular weight of SA had a significant effect on the emulsion stability.

## 3. Materials and Methods

### 3.1. Materials

Coconut milk was provided by the Taifengyuan Food Company (Haikou, China). Native SA and trifluoroacetic acid (TFA) were purchased from the Aladdin Industrial Corporation (Shanghai, China). Commercial SAs (SA-C1, SA-C2, SA-C3) were obtained from the Qingdao Mingyue Seaweed Group Co., Ltd. (Qingdao, China). Coconut milk isolate protein (CMIP) was obtained from coconut milk by centrifuging at 10,000× *g* for 10 min at room temperature [34]. All the other chemicals applied were of analytical grade.

### 3.2. Acid Treatment

SA (20 g) was dissolved in 9980 mL with stirring. TFA (7.64 mL) was added into the solution to adjust the acid concentration to 0.1mol/L [35]. The mixture was hydrolyzed in a sealed conical flask at 100 °C for 1, 2, 4, 6, and 8 h, and the precipitation was taken by static stratified centrifuge (4000× *g*, 10 min). NaHCO_3_ solution (8%, *w*/*w*) was added to the precipitate to adjust the pH to 7.5, and then anhydrous ethanol was added 4 times to obtain white flocculent precipitate which was taken by centrifugation at 4000× *g* for 10 min [36]. Ultra-pure water was used to redissolve the precipitation, and the solution was placed in a dialysis bag (3500 Da) for 12 h. SA-Ms were obtained by freeze drying.

### 3.3. Fourier Transform Infrared Spectrum (FTIR) Determination

The structure of samples before and after acid degradation was analyzed by FTIR (Avatar 370, Thermo Nicolet, USA) technology. The samples were grounded with KBr, and the spectra of the samples were obtained over the wave range of 4000–400 cm^−1^ with a resolution of 4 cm^−1^ [37].

### 3.4. Molecular Weight Determination

The molecular weight of SA was determined by gel permeation chromatography (GPC) [13], which was equipped with a Waters Dextran gel column and a Waters 2414 refractive index detector (RI). Dextran with molecular weights of 960, 1460, 7130, 12,900, 43,500, 196,000, and 401,000 Da were used as the standards. The samples dissolved in ultrapure water (3 g/L) were filtered with 0.45 μm nylon filters before injection (100 μL). NaN_3_ (0.5 g/L) was used as the eluent at a flow rate of 0.6 mL/min at the column temperature and detector temperature of 45 °C.

### 3.5. Uronic Acid Composition Determination

SA (10 mg) was suspended in 5 mL of TFA solution (2 mol/L), sealed under nitrogen atmosphere, and kept in an oven at 110 °C for 6 h. After cooling to the room temperature, 1 mL of methanol was added to the hydrolysate (1 mL). The mixture was then dried by nitrogen stream at 70 °C to remove the TFA residual. The dried hydrolysate was dissolved in 1 mL of ultrapure water and passed through a 0.22 μm aqueous membrane for subsequent analysis.

### 3.6. Rheological Measurement

The rheological property of the SA solution was carried out using a controlled temperature rheometer (Brookfield Engineering Lab., Middleboro, Mass., U.S.A.) equipped with a 60 mm parallel plate. The measurements were obtained at 25 °C with the shear rate ranged from 10 to 1000 s^−1^.

### 3.7. Interfacial Tension Measurement

Interfacial tension was determined at room temperature with the pendant drop method. A Drop Meter (A-60, Haishumai Company, Ningbo, China) was used to measure the interfacial tension between MCT and SA solutions.

### 3.8. Emulsion Preparation

CMIP (2%, *w*/*w*), medium-chain triglyceride (MCT) (5%, *w*/*w*), SA (0.6%, *w*/*w*), and sodium azide (0.04%, *w*/*w*) were mixed with ultrapure water by using high-speed emulsification dispersion at 13,000 r/min for 3 min to prepare a crude emulsion. Then, the crude emulsion was prehomogenized using an Ultra-Turrax device (24,000 rpm, 3 min) and subsequently homogenized using an ultra-high pressure homogenizer (Nano DeBEE, USA) for three passes at 50 MPa.

### 3.9. Zeta Potential Determination

The zeta potentials of SA solution (final concentration 0.01%, *w*/*v*) were determined using a Zetasizer (Nano ZS90, Malvern Instruments, Worcestershire, UK) at 25 °C. The emulsion was injected into the cuvette after 100 times dilution with the corresponding solution.

### 3.10. Interfacial Tension Measurements

The interfacial tension between the continuous phase and MCT was performed on a DropMeter (A-60, Haishumai Company, Ningbo China). Briefly, a pendant drop (bottom-to-top) of MCT was Squeeze out from a J-shaped syringe needle (needle diameter 1.500 mm) into each aqueous solution in a quartz cuvette (24 mm × 24 mm × 21 mm). The interfacial tension was calculated by fitting the drop profile with the numerical solution of the Young-Laplace equation.

### 3.11. Particle Size Determination

The particle size of emulsions was analyzed by a Malvern Mastersizer 2000 (Malvern, UK) [38]. During measurement, a few droplets of the emulsions were suspended directly in recirculating water, and the volume mean diameter *d*_4,3_ and surface mean diameter *d*_3,2_ were recorded. The refractive index and absorption index of MCT and water were set at 1.45/0.001 and 1.33/0, respectively.

### 3.12. Creaming Index (CI) Determination

CIs of the emulsions after storage at 60 °C for 0, 4, 15 and 24 days were calculated as follows:Creaming Index (%) = 100 × (HS/HT)(1)
where H_T_ represents the total height of the emulsion (cm), and H_S_ is the height of the serum layer (cm).

### 3.13. Stability Analysis of Emulsion by Multiple Light Scattering Instrument

The formed emulsions were placed in cylindrical glass tubes to study the stability of the emulsion prepared or stored at room temperature for a certain time. The stability analyzer Turbiscan adopted the principle of multi-light scattering. By measuring the changes of transmitted light and backscattering intensity with time, the concentration or particle size of the sample can be determined. In short, the relative stability of each system can be compared using TSI. The advantage of this instrument was to measure the changes of transmitted and backscattered light intensity without diluting the sample. In addition, for the emulsion samples studied, it was very sensitive to small changes in the whole emulsion system, and it was easy to monitor changes in particle size or concentration even in the early stage.

## 4. Conclusions

SAs with different molecular weight and G/M ratio were prepared by acid modification and their rheological behaviors and emulsifying properties were characterized. The structures of SA and SA-Ms were characterized by FTIR and the result showed that no significant variation of the chemical bonds was observed by acid modification among G-G, M-M, and G-M blocks in the SA chains. Meanwhile, the molecular weight of SA dropped dramatically after acid treatment. The result of the rheological analysis suggesting that the apparent viscosity of SA reduced with the increase of shear rate, and the apparent viscosity of SA-Ms dropped rapidly with the decrease of the molecular weight. The emulsifying properties of SA, SA-Ms (SA-M1, 2, 4, 6, 8) and commercial SAs (SA-C1, 2, 3) were assessed via the interface tensiometry, Turbiscan, and determination of the zeta potential, droplet size, CI, and TSI. Compared with the SA-stabilized emulsion, the interfacial tension of the emulsion stabilized by SA-M increased with the decrease of the molecular weight reduced at the similar M/G ratio. Even at neutral pH, SA can adsorb on the coconut milk isolate protein (CMIP)-coated droplet surfaces and form a self-assembled interfacial bilayer. The decrease of zeta potential and the increase of TSI of the emulsion were observed with the decrease of molecular weight, indicating that molecular weight has an important influence on the application of SA as an emulsifier.

## Figures and Tables

**Figure 1 molecules-24-04374-f001:**
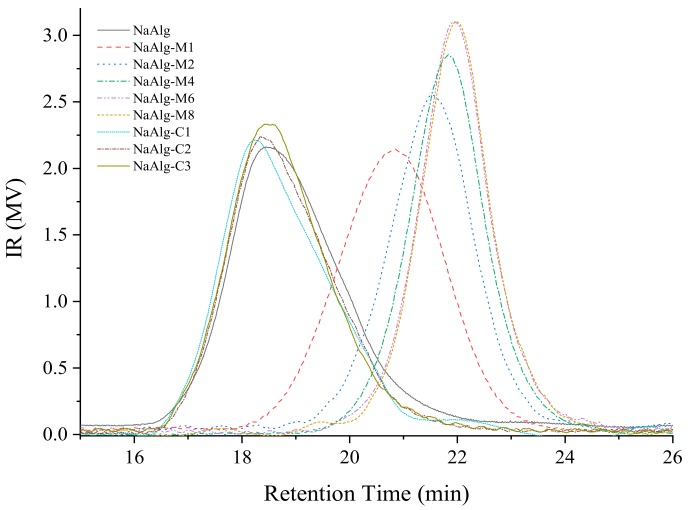
GPC profiles of SAs.

**Figure 2 molecules-24-04374-f002:**
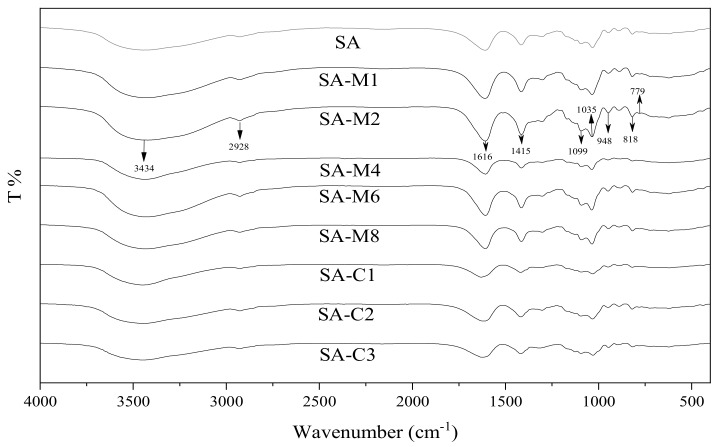
Fourier transform infrared spectrum (FTIR) of SAs.

**Figure 3 molecules-24-04374-f003:**
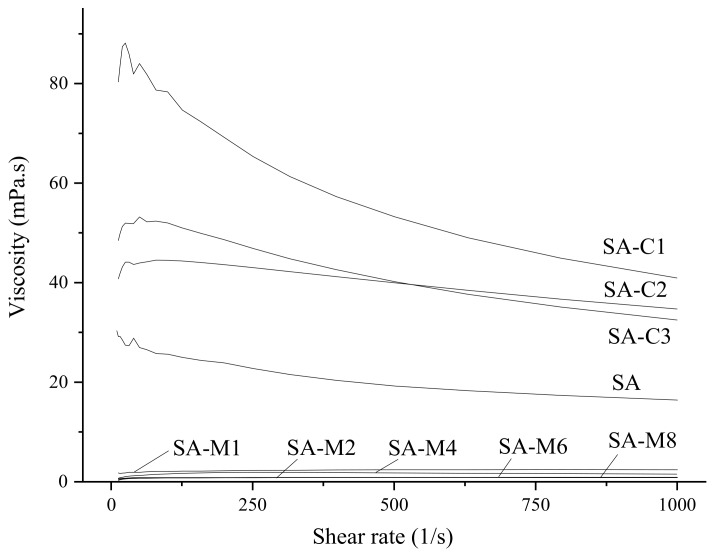
Effect of shear rate on the viscosity of SAs.

**Figure 4 molecules-24-04374-f004:**
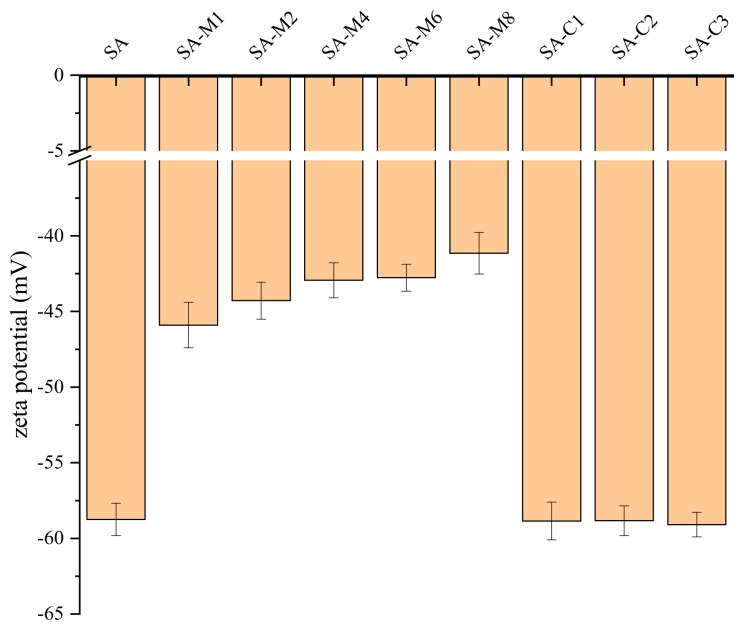
The interfacial tension of SA solution.

**Figure 5 molecules-24-04374-f005:**
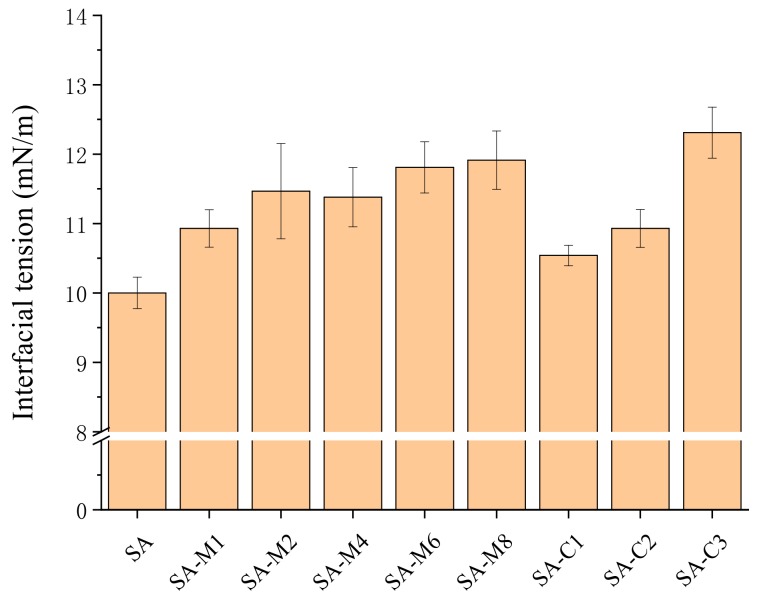
Zeta potential of the SAs-stabilized emulsions.

**Figure 6 molecules-24-04374-f006:**
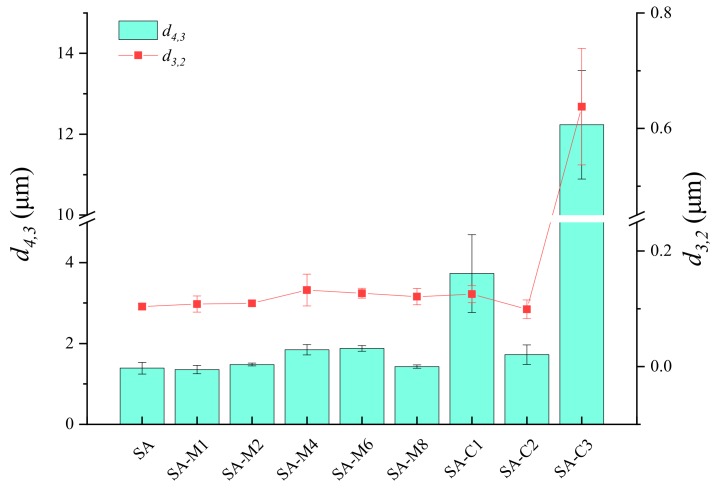
Particle sizes (*d*_4,3_ and *d*_3,2_) of the SAs-stabilized emulsions.

**Figure 7 molecules-24-04374-f007:**
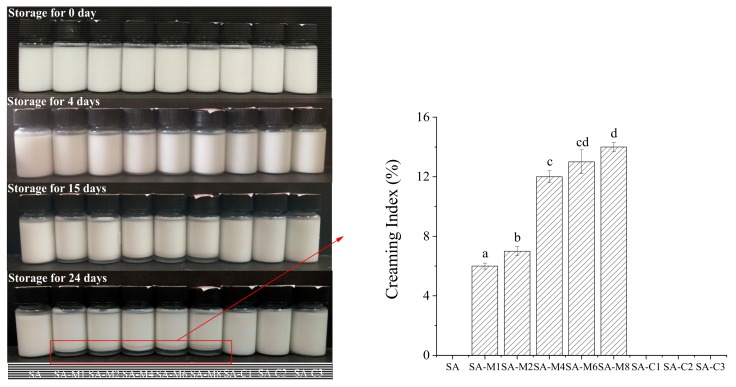
Photographs and creaming index (CI) of the SAs-stabilized emulsions during storage.

**Figure 8 molecules-24-04374-f008:**
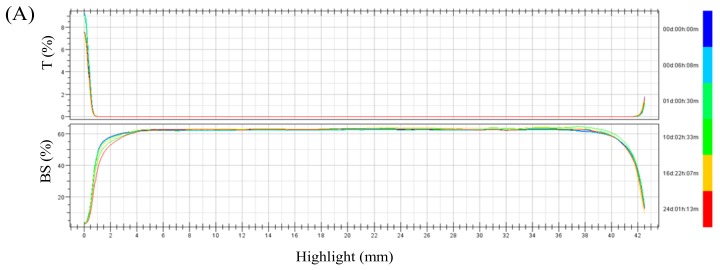
Backscattering profiles (**A**) and Turbiscan stability index (TSI) (**B**) values of emulsion during storage.

**Table 1 molecules-24-04374-t001:** Yield, molecular weight, and M/G ratio of sodium alginates (SAs) ^A^.

Sample ^B^	Yield (%) ^C^	Mw (kDa)	M (%)	G (%)	M/G
SA	100.00 ^a^	861 ± 22 ^c^	85.08 ± 0.27 ^b^	14.92 ± 0.16 ^c^	5.70 ± 0.05 ^d^
SA-M1	94.04 ± 2.82 ^b^	221 ± 11 ^d^	85.85 ± 0.41 ^a^	14.15 ± 0.22 ^d,e^	6.07 ± 0.06 ^c^
SA-M2	89.76 ± 1.31 ^c^	126 ± 6 ^e^	86.26 ± 0.33 ^a^	13.74 ± 0.12 ^e,f^	6.28 ± 0.05 ^a^
SA-M4	86.21 ± 1.66 ^d^	98 ± 5 ^f^	86.46 ± 0.46 ^a^	13.54 ± 0.13 ^f^	6.38 ± 0.06 ^a^
SA-M6	84.75 ± 0.71 ^d^	89 ± 5 ^g^	86.25 ± 0.39 ^a^	13.75 ± 0.23 ^e,f^	6.27 ± 0.06 ^a,b^
SA-M8	80.00 ± 0.98 ^e^	85 ± 8 ^h^	86.07 ± 0.21 ^a^	13.93 ± 0.16 ^e^	6.18 ± 0.04 ^b^
SA-C1	-	993 ± 46 ^a^	82.72 ± 0.45 ^c^	17.28 ± 0.16 ^b^	4.79 ± 0.06 ^f^
SA-C2	-	911 ± 24 ^b^	85.74 ± 0.30 ^a^	14.26 ± 0.13 ^d^	6.01 ± 0.05 ^c^
SA-C3	-	938 ± 33 ^a,b^	79.89 ± 0.26 ^d^	20.11 ± 0.14 ^a^	3.97 ± 0.04 ^g^

^A^ The date were reported as the mean value ± standard deviation, date with different superscript letters (a–h) in the same column were significantly different (*p* < 0.05). ^B^ SA: Sodium alginate; SA-M1-8: Sodium alginate moidfied for 1, 2, 4, 6 and 8 h; SA-C1-3: Three commercial sodium alginates. ^C^ SA-C1, SA-C2 and SA-C3 were commercial products and their yields were not obtained.

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
