# Peer review of "Effects of Molecular Weight and Guluronic Acid/Mannuronic Acid Ratio on the Rheological Behavior and Stabilizing Property of Sodium Alginate"

_molecules, 2019, doi:10.3390/molecules24234374_

Round 1

Reviewer 1 Report

The manuscript discusses the influence of alginates with various architectures on the emulsification properties, i.e., the block length and ratios of uncharged M and charged G sites. This is a very interesting and useful study, since the design of alginates has been ignored in many other investigations.

The experiments are well escribed, also some of the conclusions. The data are very useful for further research.

However, I do have difficulties with the interpretation. The stability of (oil-in-water) emulsions is defined by to different physical aspects. The emulsifier by themselves and the viscosity of the continuous water phase. Unmodified Alginates usually do not act as direct emulsifiers; they do not contain hydrophobic parts along their chains. They are not interface active. Both, polar M and charged G units dissolve in water, not in oil. Thus, the emulsifying properties need to be addressed to the viscosity effects only, as it can be seen at one of the results, that the emulsifying properties get worse with the molecular weight.

I suggest rethinking some of the interpretations and draw more intension on the viscosity effects. In addition, a direct correlation with the parameters concerning the molecular architecture, like bloc lengths, is useful to other researchers.

The manuscript discusses the influence of alginates with various architectures on the emulsification properties, i.e., the block length and ratios of uncharged M and charged G sites. This is a very interesting and useful study, since the design of alginates has been ignored in many other investigations.

The experiments are well escribed, also some of the conclusions. The data are very useful for further research.

However, I do have difficulties with the interpretation. The stability of (oil-in-water) emulsions is defined by to different physical aspects. The emulsifier by themselves and the viscosity of the continuous water phase. Unmodified Alginates usually do not act as direct emulsifiers; they do not contain hydrophobic parts along their chains. They are not interface active. Both, polar M and charged G units dissolve in water, not in oil. Thus, the emulsifying properties need to be addressed to the viscosity effects only, as it can be seen at one of the results, that the emulsifying properties get worse with the molecular weight.

I suggest rethinking some of the interpretations and draw more intension on the viscosity effects. In addition, a direct correlation with the parameters concerning the molecular architecture, like bloc lengths, is useful to other researchers.

Author Response

Response to Reviewer 1 Comments

Response: We all think well of your suggestions and the discussion about the emulsion stability has been revised. The emulsifying properties of sodium alginate, in a large part, was attributable to the viscosity effects.In addition, the molecular architecture and self-assembly form on the O/W interface will also affect the emulsifying properties of sodium alginate.

From the properties of SA-stabilized emulsion, we found the emulsion stability of SA was closely associated with the viscosity of emulsion. Higher viscosity can efficiently enhance the stability of emulsion. Combined with the results of molecular weight distribution and M/G ratio analysis, we could conclude that the molecular parameters should be the primary reasons results for the differences in the emulsifying properties of SA. SAs with higher average molecular weight (high viscosity) could improve the emulsion stability significantly. (line 166-178)

Reviewer 2 Report

Effects of molecular weight and composition sodium alginate (SA) on the rheological behavior and stabilizing properties of dispersion in emulsions were studied

I would like to make some comments that authors could take into account to improve the overall quality of the manuscript.

Comments:

Title:

In my opinion stabilizing not emulsifying properties of sodium alginate were measured. I do not see for example emulsifying activity index (EAI, m2/g) which measure emulsifying property of emulsifying agent. However, some emulsifying potential was presented (the interfacial tension of SA emulsion) but is it possible to obtain dispersion o/w using only SA?

I think that using abbreviations in the title that are not obvious to everyone is a mistake. The G and M abbreviations were used twice and only in the line 14 their meaning was explained. Generally, I recommend preparing list of abbreviations used in manuscript. However, each abbreviation that appears in the manuscript for the first time should be explained, there is no need to explain the same abbreviations repeatedly, e.g. lines 14 and 32 etc.

Table 1. The ANOVA and post comparisons should be performed for every parameter; one more column should be introduce where yields of product after hydrolysis will be presented for samples M1-M8. The abbreviations: SA, M1-M8, C1-C2 should be explain below Table. I recommend removing the molecular mass of polymer at top of peak (Mp) because this parameter does not have any practical meaning, it better to show Mz to describe fully distribution of samples.

Lines 102-104: The reduction of viscosity was observed for both samples but slope of curves was different.

Line 105: AMW = average Mw? It is necessary to be consistent during using abbreviations in manuscript.

Lines 141-143: It is over-interpretation, I do see that your conclusion is not supported by results when I compared samples SA, SA-C1/C2/C3. You have too small number of sample to generalize your observation; additionally the significant differences between samples were not proved.

Line 175: It is worth to explain correlation between TSI and stability of emulsions. does the tsi parameter have a unit? TSI is relative stability but what it mean exactly? What is the possible range for this parameter?

Lines 184-186: On which day (days) was deltaBS measured, which measurement is presented in Figure 8B?

Lines 188-190: I do not see relation between deltaBS and molecular mass of samples or their G/M ratio, etc. In my opinion this parameter poorly reflects to the stability of your samples and maybe it will be better to remove it.

Fig. 8B: DeltaBS not correlated with Fig. 7

Lines 184 and 186

Line 220-224: The relative molecular mass was determined and it is necessary to write which standards were used.

Line 240: MCT – medium-chain triglyceride? I do not find explanation in whole manuscript.

Line 274: CI?

Lines 278-279: It is over-interpretation, this relation was not proved.

Author Response

Response to Reviewer 2 Comments

Effects of molecular weight and composition sodium alginate (SA) on the rheological behavior and stabilizing properties of dispersion in emulsions were studied

I would like to make some comments that authors could take into account to improve the overall quality of the manuscript.

Comments:

1) Title: In my opinion stabilizing not emulsifying properties of sodium alginate were measured. I do not see for example emulsifying activity index (EAI, m2/g) which measure emulsifying property of emulsifying agent. However, some emulsifying potential was presented (the interfacial tension of SA emulsion) but is it possible to obtain dispersion o/w using only SA?  In addition, I think that using abbreviations in the title that are not obvious to everyone is a mistake. The G and M abbreviations were used twice and only in the line 14 their meaning was explained. Generally, I recommend preparing list of abbreviations used in manuscript. However, each abbreviation that appears in the manuscript for the first time should be explained, there is no need to explain the same abbreviations repeatedly, e.g. lines 14 and 32 etc.

Response: We all think well of your suggestions and the title has been revised according to your suggestion. The abbreviations (G/M ratio in the title) have been changed to guluronic acid/ mannuronic acid ratio. The abbreviations that appears in the manuscript for the first time have been defined. (Line 2-4)

2) Table 1. The ANOVA and post comparisons should be performed for every parameter; one more column should be introduce where yields of product after hydrolysis will be presented for samples M1-M8. The abbreviations: SA, M1-M8, C1-C2 should be explain below Table. I recommend removing the molecular mass of polymer at top of peak (Mp) because this parameter does not have any practical meaning, it better to show Mz to describe fully distribution of samples.

Response: All parameters in Table 1 were analyzed by ANOVA and the date were reported as the mean value ± standard deviation, date with different superscript letters (a–h) in the same column were significantly different (p < 0.05). And the abbreviations were defined below the Table. In addition, Mp data were deleted according to your suggestion. (line 70-74)

3) Lines 102-104: The reduction of viscosity was observed for both samples but slope of curves was different.

Response: The discussion about the slope of curves was added. From the slope of the curves, we can find that the slope fluctuation of SA-C2 was the lowest, indicating that the shear stability of SA-C2 solution was highest. Therefore, it was deduced that the viscosity of SA may be affected by both MW and M/G ratio. (line 107-110)

4) Line 105: AMW = average Mw? It is necessary to be consistent during using abbreviations in manuscript.

Response: AMW has been changed to Mw according to your suggestion. (line 110)

5) Lines 141-143: It is over-interpretation, I do see that your conclusion is not supported by results when I compared samples SA, SA-C1/C2/C3. You have too small number of sample to generalize your observation; additionally the significant differences between samples were not proved.

Response: It has been revised according to your suggestion and the significant differences have been added. Because of the limitation of samples investigated in our manuscript, the one-sided conjecture has been deleted. (line 141-145)

6) Line 175: It is worth to explain correlation between TSI and stability of emulsions. does the tsi parameter have a unit? TSI is relative stability but what it mean exactly? What is the possible range for this parameter?

Response: The stability of emulsions can be evaluated by Turbiscan stability index (TSI) parameter which takes into account all processes taking place in the sample (thickness of sediment and clear layer, process of particles settling). The higher the TSI value, the less stable the emulsion. The TSI value was calculated via the cumulative value of the backscattered or transmitted light intensity changes for each measurement compared to the previous measurement during the measurement time, reflecting the combined change in volume concentration and particle size throughout the measurement period. And it’s a ratio so the TSI does not have a unit. (line 190-196)

7) Lines 184-190: On which day (days) was deltaBS measured, which measurement is presented in Figure 8B? Lines 188-190: I do not see relation between deltaBS and molecular mass of samples or their G/M ratio, etc. In my opinion this parameter poorly reflects to the stability of your samples and maybe it will be better to remove it.

Response: We all think well of your suggestion and Figure 8B has removed. (Fig. 8)

8) Line 220-224: The relative molecular mass was determined and it is necessary to write which standards were used.

Response: Dextran standards (960 Da, 1460 Da, 7130 Da, 12900 Da, 43500 Da, 196000 Da and 401000 Da) was used in GPC determination and it has been added in the methods. (line 227-228)

Line 240: MCT – medium-chain triglyceride? I do not find explanation in whole manuscript. CI?

Response: Medium-chain triglyceride (MCT) and Creaming index (CI) have been defined when they are used for the first time in the manuscript. (line 246 and 266)

Lines 278-279: It is over-interpretation, this relation was not proved.

Response: It has been removed from the conclusion according to your suggestion. (line 281-295)

Round 2

Reviewer 1 Report

The revision improved the manuscript and in my opinion the paper can be published.

Reviewer 2 Report

Title of the manuscript is suitable. Abstract is enough informative. Materials and methods are suitable and well described. Discussion is based on the results. The paper gives to reader interesting and relevant information.

The paper has been corrected with accordance to my comments and I think that final version of this paper can be considered for publication.